# Plant Serpins: Potential Inhibitors of Serine and Cysteine Proteases with Multiple Functions

**DOI:** 10.3390/plants12203619

**Published:** 2023-10-19

**Authors:** Monaliza Macêdo Ferreira, Ariana Silva Santos, Adriadna Souza Santos, Maria Zugaib, Carlos Priminho Pirovani

**Affiliations:** 1Center for Biotechnology and Genetics, Department of Biological Sciences, Santa Cruz State University, Ilhéus 45662-900, BA, Brazil; anasilva0491@gmail.com (A.S.S.); mariazugaib@hotmail.com (M.Z.); pirovani@uesc.br (C.P.P.); 2Secretariat of Education of the State of Bahia, Salvador 41745-004, BA, Brazil; adriadna_souza@yahoo.com.br

**Keywords:** serpin, plants, defense, protease inhibitors, systematic review

## Abstract

Plant serpins are a superfamily of protein inhibitors that have been continuously studied in different species and have great biotechnological potential. However, despite ongoing studies with these inhibitors, the biological role of this family in the plant kingdom has not yet been fully clarified. In order to obtain new insights into the potential of plant serpins, this study presents the first systematic review of the topic, whose main objective was to scrutinize the published literature to increase knowledge about this superfamily. Using keywords and the eligibility criteria defined in the protocol, we selected studies from the Scopus, PubMed, and Web of Science databases. According to the eligible studies, serpins inhibit different serine and non-serine proteases from plants, animals, and pathogens, and their expression is affected by biotic and abiotic stresses. Moreover, serpins like AtSerpin1, OSP-LRS, MtSer6, AtSRP4, AtSRP5, and MtPiI4, act in resistance and are involved in stress-induced cell death in the plant. Also, the system biology analysis demonstrates that serpins are related to proteolysis control, cell regulation, pollen development, catabolism, and protein dephosphorylation. The information systematized here contributes to the design of new studies of plant serpins, especially those aimed at exploring their biotechnological potential.

## 1. Introduction

Serpins are a superfamily of protein inhibitors found in all kingdoms of life [1,2]. They are classified as irreversible inhibitors of serine proteases, mainly of the chymotrypsin class, but also inhibiting cysteine proteases [2,3].

They are highly conserved proteins, and despite some differences, all have characteristics intrinsic to the family, such as the presence of three β sheets, seven to nine α-helices, and a reactive center loop (RCL), which is a loop exposed to the solvent and the site where the reaction that leads to inhibition of target proteases occurs [4,5].

Serpins are known as suicide inhibitors due to their peculiar inhibition mechanism. Basically, the target protease recognizes the solvent-exposed serpin RCL as a potential substrate and performs a nucleophilic attack on the scissile bond P_1_–P_1’_. From that point on, the peptide bond of the RCL is broken, and a covalent acyl-ester bond is formed between the P_1_ amino acid residue and the catalytic serine of the protease. With the loop cleaved, the RCL starts to be inserted between the β A sheets of the serpin, which undergoes a conformational change in its structure and drags with it the protease linked by a covalent bond to the opposite pole of the serpin. This reaction results in partial disorder of the protease structure and distortion of its catalytic site [4,6,7], which causes the loss of proteolytic function and marks the protease-inhibitor complex for destruction [6,7].

Due to their inhibitory characteristics and different functions [6], many serpins have attracted studies of their biotechnological potential. The human genome, for example, encodes 37 serpins, many of which are fundamental for the control of various biological processes [1,7,8], such as blood clotting and inflammatory processes [7,9], tumor suppression [10], inhibition of angiogenesis [11], reduction in apoptosis [12,13], tissue remodeling, programmed cell death, blood pressure regulation, hormone transport, and molecular chaperone activity [7].

Plant serpins, in turn, have inhibitory potential against a wide range of proteases, as verified through in vitro [14,15,16] and in vivo studies [17,18]. In addition to inhibitory activities, different functions have been associated with these proteins in plant systems. Representatives such as AtSerpin1, *OSP-LRS,* and *MtSer6* from the species *Arabidopsis thaliana*, rice, and *Medicago truncatula*, respectively, are involved in defense mechanisms and cell death [17,19,20], while other serpins, such as CmPS-1 from pumpkin and BSZ4 from barley, have RNA-binding properties and β-amylase-specific molecular chaperone activity, respectively [21,22,23].

Based on their functional versatility, serpins have attracted the interest of researchers and have been found to have biotechnological potential for the production of pharmaceuticals [24,25,26,27,28] and foods, as well as for agriculture [17,29,30,31,32]. Among the examples of biotechnological potential, the human serpin antithrombin has shown antitumor and anti-angiogenesis activity [33], while Serp-1 can reduce vascular inflammatory lesions [34]. In addition, more recently, α_1_-antitrypsin was found to be able to inhibit the proteolytic activity of TMPRSS2, a serine protease with the function of activating the SARS-CoV-2 spike protein [35,36], besides preventing virus entry and replication in human cells [36].

Although to a lesser extent compared to human serpins, studies of plant serpins have identified excellent candidate proteins with potential to reduce the weight of insect pest larvae, influence the cell death caused by fungal and bacterial pathogens [17,18,29,37], and improve beer quality parameters such as wort separation rate, foam stability, viscosity, and clarity [30,31,38,39]. However, despite studies of these inhibitors, the biological role of plant serpins has not yet been fully clarified [40].

Due to the gaps in knowledge regarding the activities of serpins, many reviews have been conducted to summarize the findings about this superfamily of proteins, including their performance, inhibitory mechanisms, and biological functions [4,21,41]. However, no systematic review has been conducted considering serpins in the plant kingdom. Therefore, this work presents the first systematic review of plant serpins, describing the state of the art of research, to expand knowledge about the serpin superfamily, including their identities, localization profiles, and functions reported so far.

The review was organized into the stages of planning, execution, and summarization of data. During the initial phase, a protocol was carried out according to the PRISMA guidelines, from which the title, objectives, keywords, guiding research questions, and inclusion and exclusion criteria of studies were determined. During development, searches were performed in the selected databases using a search string, and finally, the eligible studies were summarized.

## 2. Results

The searches carried out in the Scopus, Web of Science, and PubMed databases returned a total of 533 works. The Scopus database contributed the most papers, a total of 280, followed by Web of Science with 147 and PubMed with 106 (Figure 1).

After initial analysis, 429 papers were removed. Of these, 230 were duplicates, and 199 were not in accordance with the inclusion and exclusion criteria pre-established in the protocol (Appendix A). The remaining 104 articles were submitted for full reading and evaluated for eligibility. At the end of the analysis, 90 papers were accepted for being in accordance with all the established criteria (Figure 1). All articles accepted for review are organized in Appendix A.

### 2.1. Developed Countries Invest in Studies of Plant Serpins

According to the data collected, research on plant serpins dates back to 1989. In the first two decades, research was sparse, but since 2010, research interest in these inhibitors has grown steadily (Appendix A).

The research was developed in 29 countries, of which China (16.7%), Denmark (14.44%), USA (10%), and Israel (6.7%) are the most representative, together accounting for 48% of the total studies collected. The remaining 52.17% belong to Australia and Germany, both with an average of 4.44%; France, Belgium, India, Italy, and Spain with 3.33% each; United Kingdom, Japan, Czech Republic, Ireland, Iran, and Russia with 2.22%; and Sweden, Poland, Netherlands, Romania, Lithuania, Portugal, Bulgaria, New Zealand, Tunisia, South Africa, Slovenia, and Brazil with 1.11% (Figure 2).

### 2.2. Identification and Characterization of Serpins in Different Plant Species

Serpins have been identified in many plant species, mainly in angiosperms (monocots and eudicots), as well as in mosses and algae (Appendix A).

In monocots, the serpins BSZ7, BSZx, and BSZ4 have been characterized in *Hordeum vulgare* (barley); WSZ1 (a–c) and WSZ2 (a, b) in *Triticum aestivum* (wheat); and the serpins RSZ (b–f), OSZ (a–d), *OsSRP-LRS*, and Sbser (1–3) have been described in *Secale cereale*, *Avena sativa* (oat), *Oryza sativa* (rice), and *Sorghum bicolor* (sorghum), respectively (Table 1).

In eudicots, the serpins that have been studied at the protein or gene level are mainly those from *A. thaliana* (mouse-ear cress) (AtSerpin1, *AtSRP2*, *AtSRP3*, *AtSRP4,* and *AtSRP5*). However, the serpins *MtPiI4* and *MtSer6* from *M. truncatula* and CmPS-1, CsPS-1, MdZ1b, and ApSerpin-ZX in the species *C. maxima*, *C. sativus*, *M. domestica,* and *A. praecox*, respectively, were also characterized. In algae, CrSERPIN was the only representative, characterized in the species *Chlamydomonas reinhardtii* (Table 1).

### 2.3. Biological Pathways of Plant Serpins

Protein–protein interaction (PPI) networks were constructed with the serpins that were characterized (Review Table 2) in order to trace the biological pathways to which these proteins may be related (Figure 3). Due to the differential profile observed in the localization of serpins in monocots and eudicots, two networks with only proteins selected by name and code were constructed. The networks were built with the proteins BSZx, BSZ4, BSZ7, the subfamilies WSZ1 and WSZ2, and *OsSRP-LRS*, representing the proteins from monocots, and the serpins AtSerpin1, *AtSRP2*, *AtSRP3*, *AtSRP4*, *AtSRP5,* and CmPS-1, representing the eudicots. The construction of the monocot network used *T. aestivum* as a model organism, while the eudicot network used the model organism *A. thaliana*.

The PPI network formed by serpins in monocots presented 83 proteins (nodes) and 649 connectors. Among the observed proteins, 9 are bottlenecks (betweenness values above average) and 43 are hubs (node degree values above average). Three clusters were formed with biological functions related to cell regulation (cluster 1); catabolic processes, proteolysis involved in cellular protein catabolic processes, and collagen catabolic processes (cluster 2); and pollen development (cluster 3) (Figure 3A).

The analysis with eudicot serpins formed a network with 66 proteins (nodes) and 116 connectors. Among the observed proteins, 4 are bottlenecks (betweenness values above average) and 16 are hubs (node degree values above average). Five clusters were formed with functions related to the mitotic-cell-cycle phase transition and regulation of the cell cycle (cluster 1); cysteine-type peptidase activity (cluster 2); serine-type endopeptidase inhibitor activity (cluster 3); protein dephosphorylation (cluster 4); and mitotic-cell-cycle phase transition (cluster 5) (Figure 3B).

### 2.4. Serpins Identified in Different Organs or Tissues of Plants

Among all the papers collected, 82 reported 20 serpins at the gene level or hypothetically predicted or characterized proteins that are located in different plants (Appendix A).

Serpins are ubiquitously contained in several species (Appendix A). Due to the distinct expression pattern among angiosperm species, the representation of serpin location was divided into monocots and eudicots. Of the total analyzed, 90 serpins were located in monocot species (*H. vulgare*, *T. aestivum*, *S. cereale*, *A. sativa*, *O. sativa*, *Avena fatua*, *Brachypodium distachyon*, *and A. praecox*), 25 in eudicots (*A. thaliana*, *C. maxima*, *C. sativus*, *M. domestica*, *M. truncatula*, *Centaurea maculosa*, *Citrus paradisi*, *Citrullus vulgaris*, *Pyrus communis*, *Solanum lycopersicum, and Glycine max*), and 4 in mosses (*Physcomitrella patens*). Appendix A shows the distribution of serpins in different tissues, such as grains, leaves, flowers, spikes, roots, callus, rachis, fruits, siliques, the root-shoot transition region, tissues of the abscission zone after the removal of flowers, cotyledons, stems, hypocotyls, and coleoptiles, according to the respective species (Figure 4; Appendix A).

Serpins found in monocots are mostly present in grains or seeds (56.14%). In addition, 8.42% of these proteins were identified by proteomic analysis in wheat and barley derivatives such as beer, flour, and bran (Figure 4). On the other hand, 69.69% of the serpins of eudicots are mainly distributed in leaves, roots, seedlings or shoots, flowers, and seeds (Figure 4).

It is noteworthy that serpins accumulate not only at the tissue level but also at the subcellular level, such as in the cytoplasm and nucleus of *A. thaliana* as well as in the exudate of vascular tissues (phloem) from pumpkin and cucumber (Appendix A).

### 2.5. Inhibitory Potential and the Various Targets of Plant Serpins

Plant serpins have been extensively tested in vitro to determine their inhibitory profiles against serine and cysteine-class proteases, with serine proteases being the most representative class. A total of 24 serpins with inhibitory profiles have been analyzed against 18 types of proteinases from plants, algae, mammals, insects, and bacteria (Table 2).

The proteases that have been submitted to the highest number of inhibitory assays with different serpins are, in order, chymotrypsin, trypsin, elastase, and cathepsin G, totaling 67.18% (Figure 5A).

Other proteases found to be inhibited by plant serpins (Table 2) are kallikrein; coagulation factors VIIa/sTF, Xa, and XIIa; urokinase-type plasminogen activator (u-PA); two subtilisins; thrombin; protein C; three metacaspases; a papain; and cathepsins B and L. As noted, approximately 90.64% of the inhibited proteases belong to the serine protease class (Figure 5A). Furthermore, most of the proteases tested in in vitro studies are mammalian serine proteinases (Table 2).

The serpins with the highest percentages of inhibition are BSZx from barley (19.12%), AtSerpin1 from *A. thaliana* (10.29%), and WSZ from wheat (with approximately 32.38%). Of the 17 proteinases cataloged, these serpins inhibited the activity of approximately 61.79% (Figure 5B). The other inhibitory serpins are MdZ1b, BSZ4, BSZ7, OSZ (a, b, c), RSZ (b, c1, c2, e, f), CmPS-1, Sbser (1–3), CrSERPIN, and ApSerpin-ZX (Figure 5B). Except for AtSerpin1 and CrSERPIN, these serpins mainly inhibit proteases of the serine class (Table 2).

### 2.6. Additional Functions of Plant Serpins

Changes in enzymatic activity, beer quality, allergenic effects, and interactions with RNA have all been found to be related to the action of plant serpins.

In barley, while the BSZ7 serpin negatively affected the filterability of the wort by increasing its viscosity and reducing separation rates and turbidity, the BSZ4 altered the enzymatic properties of β-amylase, increasing its maximum catalytic rate and activity during high temperatures and oxidative stress. In pumpkin and mouse ear cress, CmPS1 and AtSerpin1 interact with RNA in vitro, which may indicate a possible carrier function of phloem-mobile RNAs. Additionally, five wheat serpins (WSZ1B, WSZ1C, WSZ2A, WSZ2B, and Serpin 3) are listed as allergens because they interact with IgE antibodies from the sera of patients with wheat-associated disorders (Table 3).

### 2.7. Action of Serpins in Plant Defense and Stress Events

Plant serpins were found to be related to plant defense in 28% of the selected studies and are involved in different types of stress (Appendix A). In 25 studies, a total of 45 serpins at the gene or protein level were differentially expressed against fungal and bacterial pathogens and against different abiotic stresses (Figure 6A,B).

Among the serpins involved in biotic stresses, serpins from wheat. (Serpin 1 and WSZ1a) and barley (BSZZ4, BSZ7, and Z-type serpin) are upregulated against powdery mildew and *Fusarium* spp., respectively. In addition, 20 putative wheat serpins are also differentially expressed against the pathogens *Zymoseptoria tritici*, *Fusarium graminearum*, *Puccinia striiformis*, *Blumeria graminis,* and *Fusarium pseudograminearum* (Figure 6A) and fungal elicitors such as flg22 and chitin (Appendix A). In rice, while the OsSRP-ZXA protein is expressed in the presence of the fungus *Magnaporthe oryzae*, the *OSP-LRS* gene is upregulated upon interaction with the fungus *Rhizoctonia solani* (Figure 6A).

Similarly, two serpin-like proteins, JK86945 and JK86934, from *P. communis* are also upregulated against the fungus *Stemphylium vesicarium*. Furthermore, the expression of *MtPiI4*, *AtSRP4*, and *AtSRP5* genes from *M. truncatula* and *A. thaliana* is induced by different strains of *Pseudomonas syringae* bacteria (*Pst* DC3000, *Pst-AvrRpt2* and *Pst-AvrB*) (Figure 6A). At the protein level, AtSRP4 is also upregulated by the biocontrol bacterium *Pseudomonas fluorescens* (Appendix A).

With regard to abiotic stresses, serpins are upregulated mainly due to saline stress and water deficit, as well as by UV, methanesulfonate (MMS), osmotic and oxidative stress, temperature and humidity variations, and cryoinjury (Figure 6B).

Serpins such as *AtSRP4*, *OSP-LRS*, BSZx, and BSZ7 are expressed in response to saline stress in mouse ear cress, rice, and barley plants. In response to water deficits, *MtSer6* and *Serpin-1* levels are higher in *M. truncatula* and in a drought-tolerant variety of wheat, respectively. Furthermore, other wheat serpins, such as WSZ2a and WSZ2b, are also upregulated by water stress in different mutant strains (Figure 6B and Appendix A).

The genes *AtSRP4*, *AtSRP5*, and *OSP-LRS* are significantly expressed in *A. thaliana* and rice by ultraviolet (UV) radiation. Also in *A. thaliana*, while AtSerpin1 is induced by osmotic stress, *AtSRP2* and *AtSRP3* are significantly expressed in response to the methanesulfonate alkylating reagent (MMS) (Figure 6B and Appendix A).

Analysis of wheat varieties grown in a polytunnel system under a hot/dry or cold/humid regime during the grain filling period showed increased expression of the serpin WSZ1a. In the embryogenic callus of *A. praecox*, ApSerpin-ZX is upregulated by cold and cryopreservation, as well as by saline, oxidative, and osmotic stress (Appendix A).

The effect of overexpression or underexpression of plant serpins in response to different stresses was identified, as was the influence of these inhibitors on the cell death system in the affected tissues (Figure 7A,B).

According to the selected papers, AtSerpin1 has a protective effect on osmotic stress and is correlated with cell death during infection caused by the pathogens *Botrytis cinerea*, *Sclerotina sclerotiorum,* and *P. syringae* pv. tomato by expressing the *avrRpm1* effector and photodynamic damage by acridine orange (AO) and water stress. A cell death effect of other serpins such as *AtSRP4* and *AtSRP5* was observed with the bacterial pathogens *Pst-AvrPt2* and UV, *OSP-LRS* with *R. solani*, UV, and salt, and *MtSer6* with water deficit. In addition, overexpression of *MtPiI4* reduces bacterial populations of the strain *Pst* DC3000, and Apserpin-ZX has a positive effect on the survival rate of cryopreserved calluses of *A. praecox* (Appendix A).

In addition to the involvement of serpins in cell death caused by stress, the inhibitors Sbser1, Sbser2, and Sbser3 from sorghum, AtSerpin1 from *A. thaliana,* and CmPS-1 from pumpkin are also reported to be proteins involved in reducing the development and survival of the species *Helicoverpa zea*, *Spodoptera littoralis*, *Acyrthosiphon pisum*, and *Myzus persicae*, respectively (Appendix A).

## 3. Discussion

### 3.1. Serpins Are Ubiquitous in Different Plant Species

The presence of serpin in different organs, such as grains or seeds, may be associated with different functions, such as energy storage and defense. Serpins from wheat [16] and rye [49], for example, have glutamine repeats in their RCL, similar to prolamins, which are glutamine-rich storage proteins [16,50]. Repetition of these glutamines may serve as an attractive bait for prolamin-degrading proteinases. In addition, wheat serpins were found in the gut of *Eurygaster integriceps* (a shield bug called the sunn pest), indicating that these inhibitors are able to resist hydrolysis in the gut [59] and can act on insect digestive enzymes [14]. Thus, the presence of serpins in different plant organs is associated with defense functions against different pests and other functions not yet known.

Although serpins have been identified in different plant species, the accumulation or expression profile of these proteins has been noted mainly in grains of Poaceae family species, related not only to their functions in these tissues but also to their biological characteristics and commercial importance since crops such as wheat, barley, and rice have global importance. The data systematized here identified a higher number of serpins in these tissues.

Similarly, the pattern of expression or accumulation of serpins in tissues such as leaves, roots, siliques, flowers, and seeds of *A. thaliana* and *M. truncatula*, which are model plants, is related to the greater ease of working with small and fast-growing plants with genomes that facilitate identification of these proteins.

### 3.2. Serpins Are Promiscuous Inhibitors and Have Inhibitory Activity against Different Serine and Cysteine Proteases

The inhibitory functions of serpins have been widely observed in in vitro assays (Reverse Table 2). In general, the most widely inhibited class is the serine proteases from animals, mainly chymotrypsin (Figure 5). The strong ability that serpins have to inhibit serine proteases appears to have been inherited from a single ancestral gene [60]. However, chymotrypsin-like proteases appear to be absent in the plant kingdom. This suggests that these serpins may act by inhibiting proteases from target pathogens or that they have acquired a shift in their inhibitory profile to non-serine proteases [61].

The articles reviewed indicated that only the serpins AtSerpin1 and CrSERPIN are able to inhibit cysteine proteases. AtSerpin1 is a cross-class inhibitor, which means that it can act against different serine and cysteine proteases [29,44]. Other serpins with these characteristics are, for example, the human ov-serpins Serpin B4 and PI-9. Serpin B4 inhibits cathepsin G, human mast cell chymase, and the cysteine proteases Der p 1 and Der f 1, while PI-9 acts against different caspases and granzyme B [62]. Furthermore, within the same class, it has been observed that the BSZ, WSZ1, and WSZ2 subfamilies are able to inhibit different serine proteases [14,15,16]. This variable inhibitory capacity, so peculiar to serpins, makes them promiscuous inhibitors and true traps for different proteases [40].

Serpin promiscuity is apparently strictly related to the cleavage caused by the protease of certain residues in its loop, the RCL, the main active site of serpins [40]. This cleavage triggers events that result in protease inhibition [63,64]. A single serpin can exhibit different cleavage sites in its RCL for the same or different proteinases [4]. The serpin BSZx, for example, is cleaved by trypsin, chymotrypsin, and cathepsin G at two overlapping sites, Arg (P_1_) and Leu (P_2_), while BSZ4 is cleaved by chymotrypsin and cathepsin G at the sites Met (P_1_), Leu (P_4’_), and Lys (P_5’_) [14]. Thus, the interaction of proteases with different amino acid residues of the RCL may be an efficient strategy for serpins to act against different targets.

### 3.3. Serpins Are Multifunctional

Systems biology analysis clearly demonstrates that serpins are related to different biological pathways. They are proteins related to the control of proteolysis, phase transition and cell cycle regulation, pollen development, catabolism, and protein dephosphorylation (Figure 3A,B). In addition, proteins such as BSZ4, CmPS-1, and AtSerpin1 exhibit chaperone and RNA-binding activities. Also, some wheat serpins exhibit allergenic profiles (Table 3). Although many functions of these proteins are not well clarified, research has suggested that serpins act in the control of proteolytic activity as well as exerting functions in plant metabolism and development.

This versatility of functions appears to be a feature present in many members of this superfamily of inhibitors. In humans, many ov-serpins have multiple functions. The PAI-2 protein has an inhibitory function against urokinase-type plasminogen activator (uPA), as well as being related to defense against viral infections and having activity related to fetal development, monocyte differentiation, and metastasis [62].

This is assumed, the versatility of functions observed in human serpins also seems to be present in plant serpins. However, the multifunctional capacity of some plant serpins can be explained by the number of serpin genes present in this kingdom. While in humans and in *Drosophila melanogaster* there are 32 and 36 serpin genes, respectively [2,41], in *A. thaliana* there are 13 [2]. Possibly to compensate for the smaller number of serpin genes in the plant kingdom, these inhibitors have evolved to have multiple functions [21,40].

### 3.4. Serpins Are Defense Proteins Related to the Control of Stress-Induced Cell Death

The expression profile of serpins is affected by varied biotic and abiotic stresses (Appendix A). Thus, overexpression or underexpression of serpins such as AtSerpin1, *OSP-LRS*, *MtSer6*, *AtSRP4*, *AtSRP5*, and *MtPiI4* has an influence on plant resistance and contributes to increasing or reducing stress-induced cell death.

In *A. thaliana*, overexpression of AtSerpin1 affects development and reduces cell death caused by the fungi *B. cinerea* and *S. sclerotiorum* and the bacterium *P. syringae* pv. tomato expressing the effector *avrRpm1* [17,18]. Similarly, *AtSRP4* reduces cell death caused by *Pst-AvrRpt2*. In contrast, the knockout of *AtSRP4* and *AtSRP5* genes makes plants susceptible to cell death caused by *Pst-AvrRpt2* and UV [37].

In rice, transgenic RNAi lines with reduced expression of *OsSRP-LRS* against the fungus *R. solani*, UV and salt also lead to exaggerated cell death [19]. In *M. truncatula*, drought stress in RNAi lines with reduced *MtSer6* expression causes increased activity of a papain-like cysteine protease and premature nodule senescence [20]. Transgenic *A. thaliana* plants overexpressing *MtPiI4* are more resistant to the *Pst* DC3000 bacterial strain [65].

Studies suggest that some serpins are strictly related to cell death events. Intracellular serpins from animals, for example, act by inhibiting different proteases [66,67,68] and exert control over apoptosis. Interestingly, of the serpins cited in cell death events in this review, three are LR-type serpins, and two, AtSerpin1 and *OsSRP-LR*, are homologs of the gene *Drosophila Nec* [19], which controls cell death and is involved in the immune response of these flies [69].

The set of results observed from the published literature shows that plant serpins are defense proteins. These proteins play important roles in regulating cell death against bacteria, fungi, and abiotic stresses [17,65,70], as well as acting on insect development and survival.

## 4. Materials and Methods

The systematic review was conducted with the help of the software StArt (State of the Art through Systematic Review) v.3.3 Beta 03, developed by researchers at URFSCar (Universidade Federal de São Carlos) (http://lapes.dc.ufscar.br/tools/start_tool, accessed on 30 March 2019). The entire development of the systematic review was performed according to the PRISMA Checklist [71] and organized in three steps: planning, execution, and summarization of data.

### 4.1. Planning

During the planning phase, a protocol was developed containing the title, objectives, keywords, questions, and criteria for study selection (inclusion and exclusion) (Appendix A).

Seven questions were established to conduct the searches: I. In which countries has research with plant serpins been conducted? II. Among the serpins identified, which ones were characterized or presented more comprehensive studies, and in which species? III. In which plant parts have the serpins been identified? IV. Are plant serpins serine or cysteine protease inhibitors? And against which proteases do serpins have inhibitory potential? V. Do serpins have other functions besides the inhibitory ones? VI. Can serpins be related to defense events and plant stresses? VII. Have the biological routes and functions of serpins been well established? The construction of these questions was guided by the Population Intervention Comparison Results (PICO) guidelines [72], reducing biased responses (Appendix A).

### 4.2. Execution

The papers were identified using the search strings “serpin* AND plant*” in the PubMed, Scopus, and Web of Science databases. The resulting files were imported into BIBTEX format, compatible with the StArt (State of the Art through Systematic Review) software and submitted to the inclusion and exclusion criteria for article selection. At this stage, according to the pre-established criteria in the protocol, the articles were selected or rejected according to the extraction step.

In the extraction step, only articles that met the inclusion criteria and answered at least one of the questions were accepted. In this step, the articles were read in full, after which they were still subject to another selection criterion if they did not meet the inclusion criteria.

### 4.3. Summarization

With the eligible studies, the data from the works that answered the questions was loaded into a dynamic Excel table and summarized.

As a strategy to answer one of the questions of this review, a systems biology analysis with the sequences of some serpins present in the selected articles was performed. The sequences were obtained by direct searches in UniProt KB (https://www.uniprot.org/, accessed on 16 August 2022), using the codes or the related names. Only the proteins found through this search were used. The network was built based on the interaction of serpins and endogenous proteins.

The analysis was performed on the STRING 11.5 database (https://string-db.org/, accessed on 23 August 2022). The network was built using more than 50 interactions, a significance level of 0.7, and the addition of nodes until reaching network saturation. A file in TSV format was downloaded and analyzed using Cytoscape software version 3.6.0. To group the proteins and calculate the parameters of centrality (betweenness) and nodes (degree), the Igraph package of the R Studio statistical tool was used. Gene ontology analysis of the network clusters to determine the gene ontology (GO) categories was performed according to the categories formed by STRING 11.5.

Finally, to mitigate possible biases in the conduct of this work, the parameters of the PRISMA checklist were applied to ensure the proper description of all items presented in this systematic review (Appendix A).

## 5. Conclusions

Although considerable information has been accumulated in recent years about plant serpins, relevant gaps in the literature still exist regarding their biological roles, requiring future experimental studies. The findings reported here have provided answers regarding the localization profiles of plant serpins in different species, their inhibitory targets, and biotechnological potential described so far, and allowed the construction of a robust list of candidate proteins whose accumulation or expression levels are affected by different biotic and abiotic stresses according to proteomics and transcriptomics studies. These proteins can be targets for further research.

The reviewed results point to the broad inhibitory capacity of plant serpins for different proteases (mainly of the serine class) and their participation in the regulation of cell death caused by stress, binding to RNA, and chaperone activity, thus giving them notable biotechnological potential, especially those characterized functionally. Among the questions related to the data identified here are: What other pathogens that attack plants can be tested with these serpins? Would it be possible to exploit their potential for the treatment of human diseases? And how is their function associated with other proteins, such as those in the PPI network?

In this context, new studies are needed to evaluate the inhibitory potential of serpins not only as endogenous proteases involved in cell death but also as inhibitors of proteases in fungal and bacterial pathogens that have yet to be determined. Promising proteins such as AtSerpin1 from *Arabidopsis* and Sbser (1–3) from sorghum, whose effectiveness was observed in reducing the weight of insect pest larvae, can also be tested against new pests, or used for genetic improvement. However, despite the observed results, this superfamily of inhibitory plant proteins, which is structurally conserved in various species, has been neglected compared to the huge number of biotechnological studies of human serpins. Very few of these serpins have been tested against pathogens, and none of the serpins cited here have been evaluated for the treatment of human diseases.

Other questions, such as which endogenous or pathogenic serine proteases are affected or how these proteins act in the regulation of plant survival and defense mechanisms, still need answers. Therefore, this systematic review sheds light on the importance of the functional profile of the serpin superfamily in the plant kingdom.

## Figures and Tables

**Figure 1 plants-12-03619-f001:**
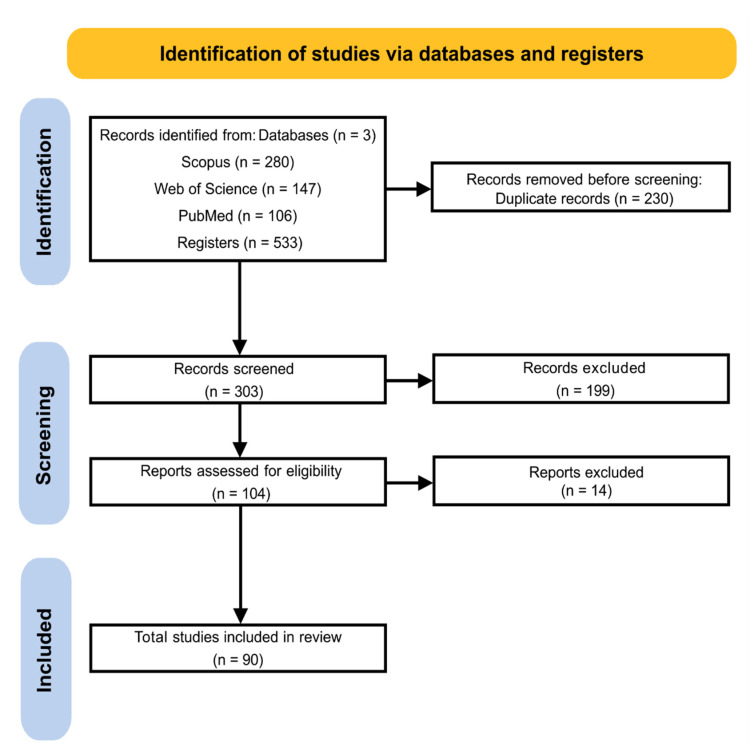
PRISMA flowchart. The flow chart shows the steps and the results in terms of the number of articles from each phase of the systematic review. The development of the flowchart was carried out according to the PRISMA guidelines [42].

**Figure 2 plants-12-03619-f002:**
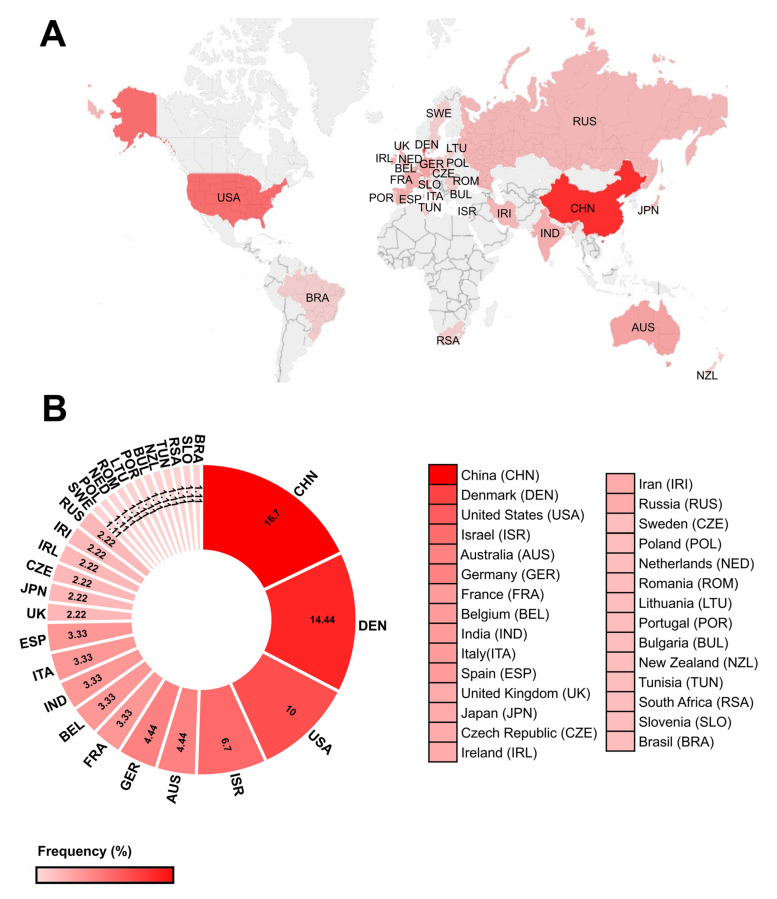
Percentage of total articles collected by country. (**A**) A physical map showing the countries that contributed to the studies selected in this systematic review. (**B**) Percentage of articles originating in each country.

**Figure 3 plants-12-03619-f003:**
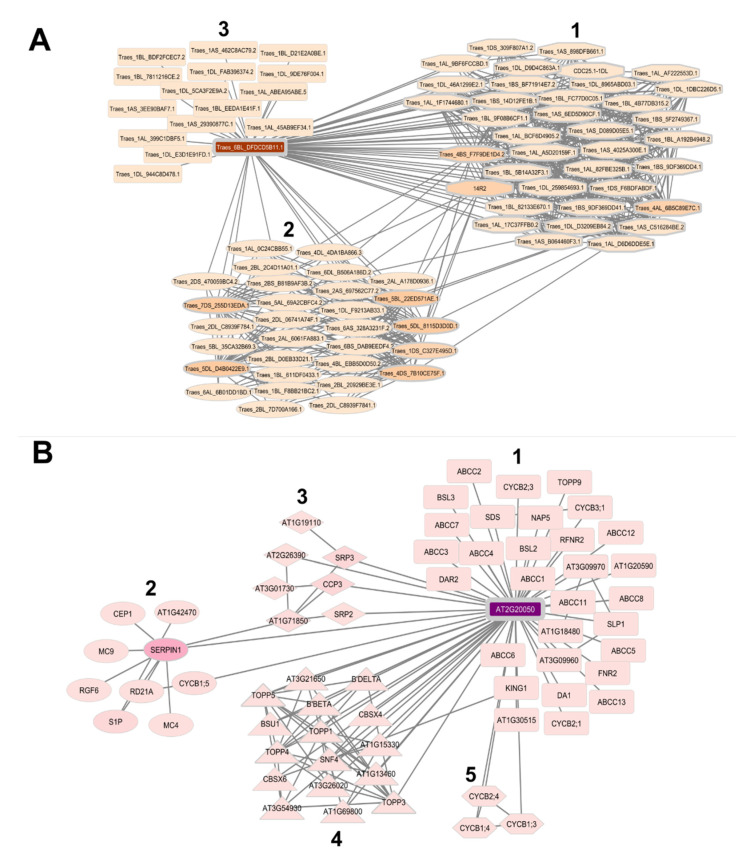
Analysis of the systems biology of plant serpins. (**A**) The network built with the serpins of monocots, presenting the functions of cell regulation (1); catabolic processes, proteolysis involved in cellular protein catabolic processes, and collagen catabolic processes (2); and pollen development (3). (**B**) The network built with eudicot serpins with functions of mitotic-cell-cycle phase transition and regulation of cell cycle (1); cysteine-type peptidase activity (2); serine-type endopeptidase inhibition activity (3); protein dephosphorylation (4); and mitotic-cell-cycle phase transition. The network was constructed using the serpins present in Table 2. The construction of the monocot network used *T. aestivum*, and the eudicot network used the model organism *A. thaliana* as a basis for the development of the network. The width of the edges of the geometric figures that contain the proteins in the networks represent the degree values (network A ≥ 1 and ≤61, network B ≥ 1 and ≤55), with the wider edges being related to the proteins with the highest value of degree, and the narrower edges refer to proteins with a lower degree value. The colors are related to the betweenness values (network A ≥ 0 and ≤1627, network B ≥ 0 and ≤1927), proteins with darker colors represent higher betweenness values and proteins with lower betweenness values are represented in lighter colors (Appendix A).

**Figure 4 plants-12-03619-f004:**
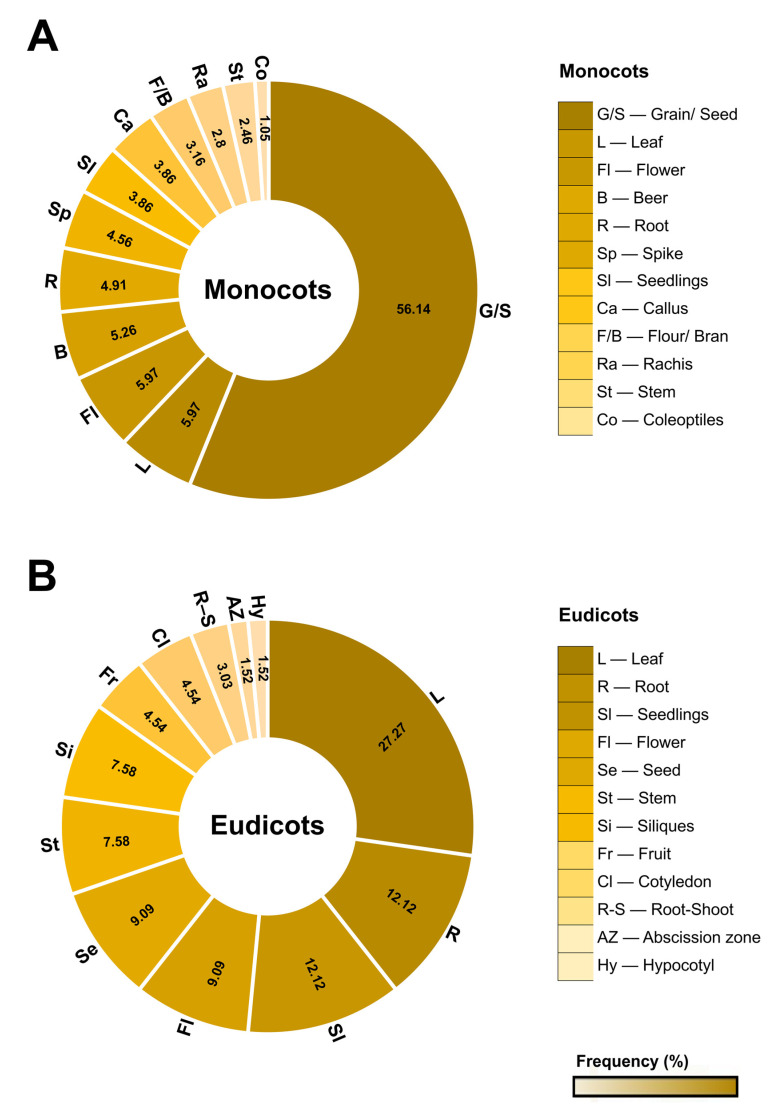
Distribution of serpins in different plant parts. Serpins are located at the protein and/or transcript level in different angiosperm plant species. (**A**,**B**) the frequency of distribution in percentage of monocots and dicots in different tissues and in products derived from wheat and barley grains.

**Figure 5 plants-12-03619-f005:**
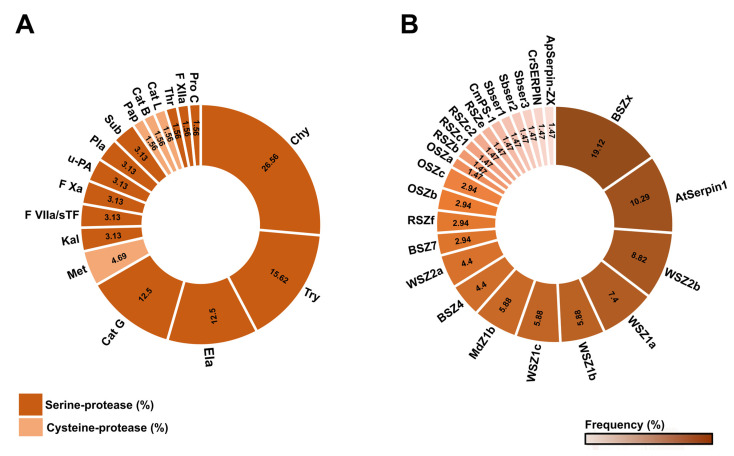
Inhibitory profile of serpins and different cognate proteases. (**A**) The identity of each protease that is inhibited to some degree by different serpins. Serine class proteases are in dark orange, and cysteine-class proteases are in light orange. Protease abbreviations are Chy (chymotrypsin), Try (trypsin), Ela (elastase), Cat G (cathepsin G), Met (metacaspase), Kal (kallikrein), F VIIa/sTF (coagulation factor VIIa/sTF), F Xa (coagulation factor Xa), u-PA (urokinase-type plasminogen activator), Pla (plasmin), Sub (subtilisin), Pap (papain), Cat B (cathepsin-B), Cat L (cathepsin-L), Thr (thrombin), F XIIa (coagulation factor XIIa), and Pro C (protein C). (**B**) The number of proteases inhibited by each serpin individually. The color pattern of the scale shows the percentage of inhibitory efficiency of serpins with different proteases.

**Figure 6 plants-12-03619-f006:**
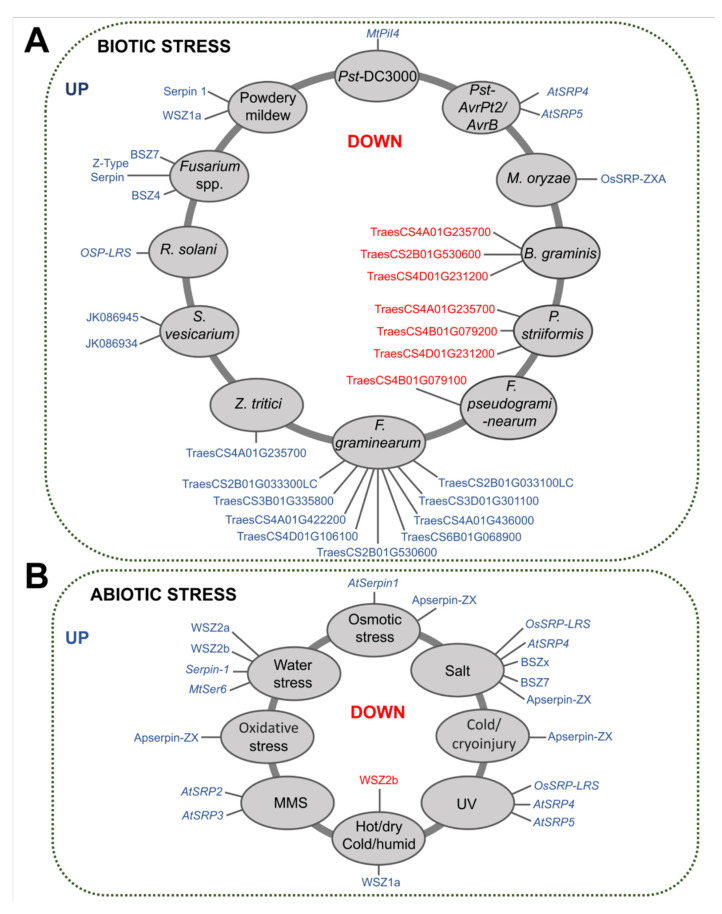
Expression pattern of plant serpins under biotic and abiotic stress. (**A**) Serpins that are upregulated or downregulated by fungal, bacterial, and insect pathogens. The circles in gray contain the microorganisms that cause the biotic stress. (**B**) Serpins that are upregulated or downregulated under different abiotic stresses. Protein expression is identified at the transcriptional and/or protein levels. Types of abiotic stresses are shown in gray circles. The set of serpins is represented by the blue outside the circles. Downregulated serpins are shown in red inside the circles. More detailed information can be found in Appendix A.

**Figure 7 plants-12-03619-f007:**
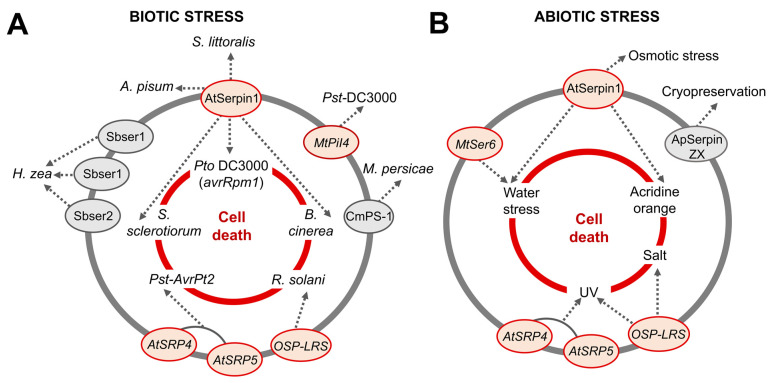
Action of serpins under different stresses. (**A**,**B**) shows the relationship of serpins in the plants’ defense systems when subjected to different biotic and abiotic stresses. Serpins in red circles are over- or under-expressed proteins that are related to cell death caused by pathogens and different abiotic stress conditions (central red circle). In gray circles are proteins that were not related to cell death, but that affected insect development (biotic stress) or had a positive effect on the survival of cryopreserved callus (abiotic stress).

**Table 1 plants-12-03619-t001:** Identity of serpins and the species in which they have been characterized.

Species		Serpin *
**Monocots**	*Hordeum vulgare*	BSZ7 (Q43492.2)
		BSZx (Q40066.1)
		BSZ4 (P06293.2)
	*Triticum aestivum*	WSZ1a (Q41593)
		WSZ1b (P93693)
		WSZ1c (Q9ST58)
		WSZ2a (Q9ST57)
		WSZ2b (P93692)
	*Secale cereale*	RSZb, RSZc1, RSZc2, RSZd, RSZe, RSZf
	*Avena sativa*	OSZa, OSZb, OSZc, OSZd
	*Oryza sativa*	*OsSRP-LRS* (Os03g41419)
	*Sorghum bicolor*	Sbser1, Sbser2, Sbser3
**Eudicots**	*Arabidopsis thaliana*	AtSerpin1 (Q9S7T8)
		*AtSRP2* (At2g14540)
		*AtSRP3* (At1g64030)
		*AtSRP4* (At2g26390)
		*AtSRP5* (At2g25240)
	*Curcubita maxima*	CmPS-1 (AAG02411.1)
	*Cucumis sativus*	CsPS-1
	*Malus domestica*	MdZ1b
	*Medicago truncatula*	*MtPiI4, MtSer6*
	*Agapanthus praecox*	ApSerpin-ZX
**Algae**	*Chlamydomonas reinhardtii*	CrSERPIN

* The papers related to these serpins and their authors can be seen in more detail in Table 2, Table 3, and Appendix A.

**Table 2 plants-12-03619-t002:** Proteases inhibited by plant serpins.

Classification	Inhibitor	Protease ^a^	Authors
Plant	AtSerpin1	Metacaspase (AtMC9)	[43]
	AtSerpin1	Papain (RD21)	[44]
			[17]
			[45]
			[46]
	AtSerpin1	Metacaspase (AtMC1)	[18]
Green alga	CrSERPIN	Metacaspase (CrMCA-I)	[47]
Animal (vertebrate)	BSZ7	Chymotrypsin, trypsin ^b^	[48]
	BSZx	Trypsin, chymotrypsin, cathepsin G	[14]
	BSZx	Plasm kallikrein, thrombin, coagulation factor VIIa/sTF, coagulation factor Xa, protein C ^b^, leukocyte elastase ^b^, coagulation factor XIIa ^b^, urokinase-type plasminogen activator (u-PA) ^b^	[15]
	BSZ4	Leukocyte cathepsin G	[14]
	BSZ4	Chymotrypsin, leukocyte cathepsin G	[22]
	WSZ1a	Chymotrypsin, cathepsin G	[14]
	WSZ1a	Chymotrypsin, cathepsin G, leukocyte elastase ^b^, pancreas elastase ^b^, coagulation factor Xa ^b^	[16]
	WSZ1b, WSZ1c	Chymotrypsin, cathepsin G, leukocyte elastase ^b^, pancreas elastase ^b^	[16]
	WSZ2a	Chymotrypsin, cathepsin G, pancreas elastase	[16]
	WSZ2b	Chymotrypsin, cathepsin G, trypsin ^b^, plasmin, plasm kallikrein, coagulation factor VIIa/sTF ^b^	[16]
	RSZb, RSZc1, RSZc2, RSZe	Chymotrypsin	[49]
	RSZf	Chymotrypsin, cathepsin G	[49]
	OSZa	Pancreas elastase	[50]
	OSZb	Pancreas elastase, chymotrypsin	[50]
	OSZc	Trypsin, chymotrypsin ^b^	[50]
	CmPS-1	Pancreas elastase	[51]
	MdZ1b	Trypsin, plasmin ^b^, chymotrypsin ^b^, urokinase-type plasminogen activator (u-PA) ^b^	[52]
	ApSerpin ZX	Trypsin	[53]
Animal (invertebrate)	AtSerpin1	Trypsin, chymotrypsin, cathepsin B and L	[29]
	Sbser1, Sbser2, Sbser3	Trypsin	[32]
Bacterium	BSZx	Subtilisin Carlsberg and Novo ^b^	[15]

^a^ Data on plant serpins and cognate proteases were computed from in vitro and/or in vivo experimental data. ^b^ Weakly inhibited proteases.

**Table 3 plants-12-03619-t003:** Serpins with non-inhibitory functions.

Other Conditions		
Serpins	Description	Authors
WSZ2b, WSZ1c	Serpins were identified as wheat allergens because they interacted with IgE antibodies from the sera of patients with allergies to wheat, identified by electrophoresis and IgE-immunoblotting analysis.	[54]
WSZ1b		[54,55]
WSZ2a	WSZ2A was identified among the IgE-binding proteins of German bakers with work-related wheat flour allergies (asthma/rhinitis).	[56]
Serpin 3	The serpins reacted with the IgG and IgA antibodies of patients with celiac disease or dermatitis herpetiformis. The antibody reactivity to non-gluten antigens was further confirmed with a recombinant serpin, Serpin 3 (gi: 224589270), which showed reactivity with IgG and/or IgA antibodies from the patients.	[57]
BSZ7	In particular, when the serpin Z7 was added to the wort, changes in its filterability were observed, such as a slight increase in viscosity and a reduction in the separation rate and turbidity, which may affect the final quality of the beer.	[30]
BSZ4	The presence of serpin Z4 changed β-amylase enzymatic properties by increasing the maximal catalytic velocity and stabilizing β-amylase activity during rising temperature and oxidative stress.	[22]
CmPS-1	*C. maxima* phloem serpin 1 (CmPS1) is able to bind RNA in vitro, and among the different RNAs tested, it has the highest affinity for tRNA in plants.	[23,58]
AtSerpin1	Similar to CmPS-1, AtSerpin1 formed complexes with tRNA from yeast and *A. thaliana* leaves.	[58]

## Data Availability

The data presented in this study are available in the article/Appendix A.

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
