# Peer review of "Plant Serpins: Potential Inhibitors of Serine and Cysteine Proteases with Multiple Functions"

_plants, 2023, doi:10.3390/plants12203619_

Round 1
Reviewer 1 Report
Plant serpins, potential inhibitors of serine and cysteine-proteases with multiple functions
By Monaliza Macêdo Ferreira, Ariana Silva Santos, Adriadna Souza Santos, Maria Zugaib, Carlos Priminho Pirovani
Discuss Plant serpins as potential inhibitors of serine and cysteine-proteases and their emerging importance in the present time. The authors provide a comprehensive account of plant serpins, characterization in different plant species, inhibitory functions, and other biological activities.
How can the systemized information in this review contribute to new studies on plant serpins? Explain. What is the need for this kind of literature review? Discuss.
In the introduction, information on Serpins is randomly presented, it needs to be rearranged for clarity and consistency.
What is the biotechnological importance of Serpins, please explain in the introduction section.
Line 77-line 87: This portion on systematic and narrative reviews description is not required and relevant in the present context. Kindly remove it.
Chapter 2.7. The biological pathways of plant serpins should be discussed before their biological functions. Please rearrange the literature.
Figures are well-drawn and nicely presented, kindly improve the resolution for better understanding.
Conclusion part- needs to be written precisely discussing plant serpins and their emerging biotechnological importance and how this article provides/increase knowledge on plant serpins. Reformulate this part.
Minor comments:
Line 20: Based on this, were selected and summarized data….reformulate this sentence, the meaning is not clear.
English language needs to be improved, the paper contains many grammatical mistakes.
References: I could not see many new studies in the reference section. Most of the cited papers are old.
Please add the current studies/information on serpins/plant serpins to improve the quality of the manuscript. Please revise.
Extensive English revision of the paper is needed before consideration in the journal.
Author Response
Editors of Plants
We sincerely appreciate the contribution in the editorial review and suggestions regarding the manuscript " Plant serpins, potential inhibitors of serine and cysteine-proteases with multiple functions." (Manuscript ID: plants-2426564)
All suggestions from Reviewers #1 and Reviewers #2 were entirely accepted and highlighted in the text.
Please find our comments on reviewers' suggestions below.
Sinceriously yours,
Monaliza Macêdo Ferreira
Reviewers #1
- How can the systemized information in this review contribute to new studies on plant serpins? Explain. What is the need for this kind of literature review? Discuss - We appreciate your comment. We have added new information has been added in section 5 (Conclusion) to answer these questions.
- In the introduction, information on Serpins is randomly presented, it needs to be rearranged for clarity and consistency - The introduction has been rewritten and reorganized for better understanding.
- What is the biotechnological importance of Serpins, please explain in the introduction section - Information on the biotechnological importance of serpins has been added in the introduction lines 71 – 85.
- Line 77-line 87: This portion on systematic and narrative reviews description is not required and relevant in the present context. Kindly remove it - Information has been removed accordingly.
- Chapter 2.7. The biological pathways of plant serpins should be discussed before their biological functions. Please rearrange the literature - Section reorganized accordingly.
- Figures are well-drawn and nicely presented, kindly improve the resolution for better understanding - The quality of all figures has been increased to 600 DPI.
- Conclusion part- needs to be written precisely discussing plant serpins and their emerging biotechnological importance and how this article provides/increase knowledge on plant serpins. Reformulate this part - We appreciate your feedback. The conclusion of the manuscript has been redrafted. Lines 649 – 678.
Minor comments:
- Line 20: Based on this, were selected and summarized data….reformulate this sentence, the meaning is not clear - The sentence has been rewritten for better understanding of the reader. Lines 20 – 21.
- English language needs to be improved, the paper contains many grammatical mistakes - The English language of the entire manuscript was proofread by a native speaker.
- References: I could not see many new studies in the reference section. Most of the cited papers are old - Current references have been included in the text and consequently in the reference list. Line 731.
- Please add the current studies/information on serpins/plant serpins to improve the quality of the manuscript. Please revise - We added one more recent study to improve quality. Line 584. However, much that is known about plant serpins is present in the summarized studies and in what was discussed.
- Comments on the Quality of English Language: Extensive English revision of the paper is needed before consideration in the journal - The English language of the entire manuscript was proofread by a native speaker.
Reviewer 2 Report
This manuscript promises a thorough analysis of the roles of plant serpins to serve as a foundation for future studies. It does not disappoint, but utilizes 90 studies to describe the different genes, expression patterns and targets of serpins. The manuscript does exactly as offered, but as a meta-analysis, it is not novel. Plant serpins are a relatively minor field, so the significance of content and interest to readers is low. Since the study does not offer any hypotheses and compare the positive or negative results, the scientific soundness is not extraordinary. The delightful figures and systematic approach are high quality.
Major Revisions:
Line 165-166 and elsewhere: there is a publication bias in regards to what plants will have the most thorough testing (model species). I doubt that the "tissues of the abscission zone after the removal of flowers" were tested for Secale cereale as much as for Arabidopsis thaliana. The bias in publication needs to be addressed as a caveat for the study.
Figure 7 is too busy and could be excluded.
Section 4.1 should be moved to the end of the introduction to better set up your study and let the reader know where you are going.
Minor edits:
Line 23: "related" not needed
Lines 77-87: irrelevant, delete
Line 105 and Methods: describe these criteria better
Lines 231, 243 and Figure 3: The discovery of serpins in beer needs better introduction and explanation
Lines 356-373: Can be deleted
Line 430: "In view of the results". This is assumed.
Line 445: decapitalize cinerea
Author Response
Editors of Plants
We sincerely appreciate the contribution in the editorial review and suggestions regarding the manuscript " Plant serpins, potential inhibitors of serine and cysteine-proteases with multiple functions." (Manuscript ID: plants-2426564)
All suggestions from Reviewers #1 and Reviewers #2 were entirely accepted and highlighted in the text.
Please find our comments on reviewers' suggestions below.
Sinceriously yours,
Monaliza Macêdo Ferreira
Reviewers #2
- Line 165-166 and elsewhere: there is a publication bias in regards to what plants will have the most thorough testing (model species). I doubt that the "tissues of the abscission zone after the removal of flowers" were tested for Secale cereale as much as for Arabidopsis thaliana. The bias in publication needs to be addressed as a caveat for the study - We appreciate your feedback. Some excerpts have been modified to improve understanding. Lines 207 - 212; 260 - 270; 286 - 287. We also deal with the subject in the Discussion Section. Lines 494 - 503.
- Figure 7 is too busy and could be excluded - We appreciate your comment. However, it was not clear to us why figure 7 (relocated as figure 3) should be deleted. Figure 7 was elaborated with the eligible studies in which the serpins were characterized and using the Arabidopsis and wheat species for the construction of biological pathways in monocots and eudicots. At this point we want to understand the interaction of serpins with other proteins and what functions may be related to them for comparison with the literature. Therefore, we consider a figure that summarizes important information about new targets or related pathways in these inhibitors. Furthermore, it is not uncommon for systematic review studies to use the PPI network to enhance the value of the sitematized data. I cite here a recent paper by Santos et al., 2023 that used the network to make better use of the systematized data (Title: State of the Art of the Molecular Biology of the Interaction between Cocoa and Witches' Broom Disease: A Systematic Review).
- Section 4.1 should be moved to the end of the introduction to better set up your study and let the reader know where you are going - Section 4.1 and part of the review methodology has been added briefly in the final paragraph of the introduction. Lines 93 - 98.
Minor edits:
- Line 23: "related" not needed - Have been corrected and revised accordingly. Line 25.
- Lines 77-87: irrelevant, delete - Have been corrected and revised accordingly.
- Line 105 and Methods: describe these criteria better - Have been corrected and revised accordingly. Lines 164 – 165.
- Lines 231, 243 and Figure 3: The discovery of serpins in beer needs better introduction and explanation - Have been corrected and revised accordingly. Lines 83 – 84; 338 – 339 and Table 3.
- Lines 356-373: Can be deleted - Have been corrected and revised accordingly.
- Line 430: "In view of the results". This is assumed. - Have been corrected and revised accordingly. Line 551.
- Line 445: decapitalize cinerea - Have been corrected and revised accordingly. Line 566.